# *PALB2* Variants Extend the Mutational Profile of Hungarian Patients with Breast and Ovarian Cancer

**DOI:** 10.3390/cancers15174350

**Published:** 2023-08-31

**Authors:** Henriett Butz, Petra Nagy, János Papp, Anikó Bozsik, Vince Kornél Grolmusz, Tímea Pócza, Edit Oláh, Attila Patócs

**Affiliations:** 1Department of Molecular Genetics, The National Tumor Biology Laboratory, National Institute of Oncology, Comprehensive Cancer Center, 1122 Budapest, Hungarybozsik.aniko@oncol.hu (A.B.); grolmusz.vince@oncol.hu (V.K.G.); pocza.timea@oncol.hu (T.P.); olah.edit@oncol.hu (E.O.); patocs.attila@oncol.hu (A.P.); 2Department of Oncology Biobank, National Institute of Oncology, 1122 Budapest, Hungary; 3Hereditary Tumours Research Group, Eötvös Loránd Research Network, 1089 Budapest, Hungary; 4Department of Laboratory Medicine, Semmelweis University, 1092 Budapest, Hungary

**Keywords:** *PALB2*, genetics, breast cancer, hereditary, genetic predisposition, genetic testing, next-generation sequencing, ovarian cancer

## Abstract

**Simple Summary:**

*PALB2* is the third most important breast cancer susceptibility gene after *BRCA1* and *BRCA2*, presenting with varying prevalence and mutational profiles in different populations. We prospectively evaluated the prevalence of germline *PALB2* genetic variants in 1848 (1280 breast and 568 non-breast) consecutive Hungarian cancer patients between 2021 September and 2023 March. In addition, 191 young (<33 years, yBC) breast cancer cases were also tested. These data were compared with data of 134,187 non-cancer individuals retrieved from the Genome Aggregation Database. Twenty-one breast cancer (1.4%) and one non-breast cancer patient (0.17%) carried pathogenic/likely pathogenic *PALB2* variants. One particular variant (NM_024675.4:c.509_510delGA) was relatively common, presented in one-third of the cases among Hungarian patients with *PALB2* variants. Including *PALB2* in the routine molecular genetic testing of breast cancer patients is recommended because it is associated with high cancer risk, and preventive and screening programs in *PALB2* carriers may improve their life expectancy similarly to *BRCA1/2* carriers.

**Abstract:**

Background: The pathogenic/likely pathogenic (P/LP) variant detection rate and profile of *PALB2*, the third most important breast cancer gene, may vary between different populations. Methods: *PALB2* was analyzed in peripheral blood samples of three independent cohorts: prospectively between September 2021 and March 2023 (i) in 1280 consecutive patients with breast and/or ovarian cancer (HBOC), (ii) in 568 patients with other cancers (controls), and retrospectively, (iii) in 191 young breast cancer (<33 years, yBC) patients. These data were compared with data of 134,187 non-cancer individuals retrieved from the Genome Aggregation Database. Results: Altogether, 235 cases (235/1280; 18.3%) carried at least one P/LP variant in one of the HBOC susceptibility genes. P/LP *PALB2* variants were identified in 18 patients (1.4%; 18/1280) in the HBOC and 3 cases (1.5%; 3/191) in the yBC group. In the control group, only one patient had a disease-causing *PALB2* variant (0.17%; 1/568) as a secondary finding not related to the disease, which was similar (0.15%; 205/134,187) in the non-cancer control group. The NM_024675.4:c.509_510delGA variant was the most common among our patients (33%; 6/18). We did not find a significant difference in the incidence of *PALB2* disease-causing variants according to age; however, the median age of tumor onset was lower in *PALB2* P/LP carriers versus wild-type patients (44 vs. 48 years). In our cohort, the odds ratio for breast cancer risk in women with *PALB2* P/LP variants was between 8.1 and 9.3 compared to non-HBOC cancer patients and the non-cancer population, respectively. Conclusions: *PALB2* P/LP variants are not uncommon among breast and/or ovarian cancer patients. Their incidence was the same in the two breast cancer cohorts studied but may occur rarely in patients with non-breast/ovarian cancer. The c.509_510delGA variant is particularly common in the studied Hungarian patient population.

## 1. Introduction

The PALB2 protein functions as the BRCA2 partner and localizer, and it is necessary for homologous DNA recombination (HR) to repair double-strand DNA breaks. For this repair process, the BRCA1 and BRCA2 molecules must be brought together by the PALB2 protein, which serves as a link between them [1,2]. The formed BRCA1-PALB2-BRCA2 “tri-molecular complex” is an important component of the HR repair system that provides high-fidelity, template-dependent repair of complex DNA damages [3,4].

The increased risk of germline *PALB2* pathogenic/likely pathogenic (P/LP) variants for breast cancer was initially suggested in 2007, and it was regarded as the third most significant breast cancer gene after *BRCA1* and *BRCA2* following the publication by Antoniou et al. about its breast cancer risk estimate that seemed to overlap with *BRCA2* [5,6]. *PALB2* P/LP variants represent an increased risk for female breast cancer that falls between the classic “high” and “moderate” categories. The estimated risk of P/LP *PALB2* variants for female breast cancer was determined as 53% [6,7], while only a modestly increased risk for ovarian cancer was implied (4.8% to age 80) [8]. Interestingly, in the case of the *PALB2* gene, both risk estimates are strongly influenced by family history, and the estimated absolute risk of developing cancer by age 80 years varies from 52% to 76% and 5% to 16% regarding breast and ovarian cancer, respectively, depending on the presence of familial presentation [3,9]. Based on these risk estimates, both the National Comprehensive Cancer Network (NCCN) and the American College of Medical Genetics and Genomics (ACMG) recommend surveillance protocols for *PALB2* P/LP variant carriers, similar to *BRCA1/2* P/LP variant carriers [3,9]. Annual mammography beginning at age 30 is endorsed, and breast MRI screening may also be considered. The role of risk-reducing mastectomy has been waiting to be determined but may be considered. Risk-reducing salpingo-oophorectomy may also be advisable in carriers at age > 45 years [3,9]. By these approaches, *PALB2* P/LP carriers can already benefit from the advantages of primary and secondary prevention.

Overall, studies suggest that patients with breast cancer harbor P/LP *PALB2* variants in 0.4–3% of the cases; however, it is also suggested that the prevalence strongly varies in different populations [4,9]. Additionally, while some initial studies assumed associations between a *PALB2* P/LP variant and increased risk of triple-negative breast cancer [10,11,12], it is now suggested that there is no established genotype–phenotype correlation, which also can be explained by the different characteristics of different populations [3].

Data on heterozygote *PALB2* disease-causing variant carriers compared to *BRCA1/BRCA2* carriers are still scarce in terms of both cancer incidence, spectrum of cancers and clinical outcomes [6]. The frequency of P/LP *PALB2* varies, hence their clinical significance can differ among different populations [4]. Therefore, we aimed to investigate the *PALB2*/LP variant prevalence and mutational spectrum in the Hungarian HBOC patients, including 191 very young cases, to compare it to patients with non-HBOC tumor types and to a healthy, non-cancer control population. We also assessed the potential effect of variants of uncertain significance and genotype–phenotype associations.

## 2. Materials and Methods

### 2.1. Subjects

*PALB2* variants were analyzed in three independent cancer patient cohorts (Table 1). Patients were referred for molecular genetic testing at our national center (Department of Molecular Genetics, Comprehensive Cancer Center, National Institute of Oncology) by clinical geneticists.

The first two cohorts consisted of 1280 consecutive cancer patients with breast and/or ovarian cancer (HBOC) and 568 patients presenting with other cancers (as cancer controls), prospectively investigated between September 2021 and March 2023, regardless of their gender or age. Indication of genetic testing was established following current NCCN guideline (NCCN Clinical Practice Guidelines Genetic/Familial High-Risk Assessment: Breast, Ovarian and Pancreatic and the Hungarian Ministry of Human Resources’ professional healthcare guideline on genetic counseling (No. 20 of 2020. EüK., effective from 14 December 2020. http://www.hbcs.hu/uploads/jogszabaly/3278/fajlok/2020_EuK_20_szam_EMMI_szakmai_iranyelv_2.pdf (accessed on 1 September 2021)) [13]. Their genetic analysis was performed using a multigene panel within the routine clinical genetic care. The third, independent patient cohort was represented by 191 young breast cancer (<33 years, yBC) patients, assessed retrospectively.

According to Hungarian legal and ethical regulations, germline genetic analysis was performed following genetic counseling. Each patient gave informed consent to the genetic test based on the approval of the Scientific and Research Committee of the Medical Research Council of the Ministry of Health, Hungary (ETT-TUKEB 53720-4/2019/EÜIG, ETT-TUKEB 4457/2012/EKU).

To compare allele frequencies, population data from the Genome Aggregation Database (gnomAD v.2.1.1) was used applying the European non-Finnish non-cancer population (n = 134,187) (accessed on 3 July 2023) [14].

### 2.2. Genetic Analysis

DNA extraction from peripheral blood was done by Gentra Puregene Blood Kit (#158389, Qiagen, Hilden, Germany), as previously reported [15]. Mutational profile and copy number analysis were performed using the TruSight Hereditary Cancer Panel version 2.0 (#20029551, Illumina, San Diego, CA, USA). Sequencing was run on an Illumina MiSeq instrument with MiSeq Reagent Kit v3 (600 cycles) (#MS-102-2002, Illumina, San Diego, CA, USA). All pathogenic/likely pathogenic and *PALB2* variants of uncertain significance were validated on a second independently extracted DNA sample by conventional Sanger sequencing and multiplex ligation-dependent probe amplification (SALSA MLPA Probemix P260 PALB2-RAD50-RAD51C-RAD51D, MRC-Holland, Amsterdam, The Netherlands). Sanger sequencing and MLPA showed 100% concordant results with the NGS method.

### 2.3. Data Analysis and Variant Classification

NGS data were analyzed by the Illumina Dragen Enrichment pipeline (v.4.0.3, San Diego, CA, USA), where both sequence variants and copy number alterations were assessed. In nucleotide detection, correct nucleotide reads with a higher than Phred quality score of 30 were accepted per position. GRCh37 genome build and MANE Select transcripts were used as reference sequences. In mapping metrics, input read number was on average 2.5 million/sample, and average % of proper reads was 95. In variant calling, variant allele frequency (VAF) between 30 and 70% was accepted for heterozygosity. Average base coverage for HBOC genes was 208 (min: 76, max: 594 reads/base). Low covered bases (<10 reads/bp) represented an average of 0.1% per gene. Variants were classified following the guidelines of the ACMG [16] and were cross-checked in the BRCA Exchange (https://brcaexchange.org/ (accessed between 1 September 2021 and 1 June 2023)), NCBI ClinVar (https://www.ncbi.nlm.nih.gov/clinvar/ (accessed between 1 September 2021 and 1 June 2023)), NCBI ClinGen (https://www.clinicalgenome.org/ (accessed between 1 September 2021 and 1 June 2023)), Varsome (https://varsome.com/ (accessed on)), and Franklin (https://franklin.genoox.com/clinical-db/home (accessed between 1 September 2021 and 1 June 2023)) databases. Variant interpretation and cross-referencing in different databases were consecutive during patient care and were accessed between September 2021 and July 2023.

### 2.4. Statistical Analyses

Statistics were carried out by GraphPad QuickCalcs (https://www.graphpad.com/quickcalcs/ (accessed on 12 July 2023)) and MedCalc (https://www.medcalc.org/calc/comparison_of_proportions.php (accessed on 12 July 2023)). Depending on sample size, a two-sided Fisher exact test or an “N-1” Chi-squared test was used to compare allele frequencies between cases and population controls and to calculate 95% CIs. Age of onset curves were compared using the log-rank (Mantle–Cox) test. Results were considered statistically significant when *p* < 0.05.

## 3. Results

### 3.1. PALB2 Detection Ratio and Mutational Profile in Breast Cancer Patients

We assessed the frequency of *PALB2* variants in two breast cancer cohorts and a non-HBOC cancer patient cohort (Table 1).

Of the 1280 consecutive HBOC patients, disease-causing variants in at least one of the breast cancer susceptibility genes were detected in 235 cases (235/1280; 18.3%). *PALB2* genetic variants were identified in 58 cases (P/LP in 18 and VUS in 40 cases) (Figure 1a). In the 18 HBOC patients, 11 different P/LP *PALB2* variants were detected (Table 2). The *PALB2* P/LP ratio in this cohort was 1.4% (18/1280). By comparing this finding to other breast cancer susceptibility genes recommended to be tested by the NCCN guideline [9,13], we found that the P/LP ratio in the *PALB2* gene was the fourth most prevalent. As expected, *BRCA1* was the most prevalent (68/1280; 5.3%), followed by *BRCA2* (62/1280; 4.8%) and *CHEK2* (26/1280; 2%). Among the consecutively referred breast cancer patients, 43 were males (vs. 1237 females). The *PALB2* P/LP variant was detected only in one of them. This detection ratio was similar in male (1/43; 2.3%) and female (21/1237; 1.6%) breast cancer patients. “Double mutation” (P/LP variant in more than one HBOC susceptibility gene) was observed in 13 cases; however, none of them involved any *PALB2* disease-causing genetic alteration.

By analyzing genotype–phenotype associations, we found histological characteristics of the tumors of *PALB2* P/LP carrier patients similar to *BRCA2* carriers (Figure 1b). Also, age of first tumor onset and Ki67 proliferation indices in patients with the P/LP *PALB2* variant were similar to *BRCA2*-associated tumor patients (Figure 1b), and no difference in multiplex tumor occurrence among *BRCA1, BRCA2*, or *PALB2* carriers were observed.

However, the median age of tumor onset was 44 years in *PALB2* P/LP carriers versus wild-type patients where it was 48 years (*p* = 0.0503, Figure 1c)**.**

Regarding *PALB2* VUSs, we identified 40 patients carrying 24 different *PALB2* VUSs (Table 2). Out of the 235 positive HBOC patients, 9 carried a *PALB2* variant of uncertain significance (VUS) in addition to a P/LP variant in any of the other HBOC-associated genes. To assess the potential association of *PALB2* VUSs, we compared clinicopathological parameters of genetically wild-type, *PALB2* P/LP, and VUS carriers. Triple-negative histology was 40% among P/LP carriers compared to 20% of the patients with wild-type and VUS carriers; however, this difference did not reach statistical significance. Regarding multiplex tumor occurrence, estrogen positivity, and HER2 positivity, no differences among the three groups with different genotypes were detected (Figure 1b).

Among the 191 young breast cancer patients, P/LP *PALB2* variants were identified in three cases (1.5%; 3/191), which did not differ from the detection ratio observed in the whole HBOC group (Table 2).

### 3.2. Pathogenic/Likely Pathogenic PALB2 Variant Detection Ratio in Non-HBOC Oncology Patients and Non-Cancer Control Population

Incidental findings of the *PALB2* P/LP variant in a patient with a non-HBOC phenotype indicate the extent of the penetrance. Therefore, we compared this in oncological and population control cohorts.

In the oncological control group—patients with non-HBOC tumors—only one patient had a disease-causing *PALB2* variant (0.17%; 1/568) as a secondary finding not related to the disease (Table 2). This particular patient had endometriosis at the age of 34 years. The genetic testing was indicated in her case because her mother and grandmother had breast cancer at the age of 45 and 60 years, respectively; however, both of them were unavailable for genetic testing. Due to the young age of this *PALB2* P/LP variant carrier, *PALB2*-related tumor types could not be excluded in the future.

Expectedly, HBOC patients had a higher risk of detecting *PALB2* P/LP variants compared to non-HBOC patients and the non-cancer population with OR 8.1 and 9.3, respectively (Table 3).

We compared the *PALB2* P/LP detection ratio, as an incidental finding, to the high-penetrance *BRCA1* and *BRCA2* genes among non-HBOC patients and found similar frequencies (Table 4). While the moderate-penetrance *ATM* and *CHEK2* disease-causing variants were more frequent as secondary findings, 0.35% and 0.44%, respectively, the difference was not statistically significant compared to *PALB2* (0.088%), (*p* = 0.3739 and *p* = 0.2175, respectively).

Additionally, we assessed the frequency of *PALB2* P/LP variants in the gnomAD non-cancer cohort, which showed similar detection ratios in comparison to our cancer control groups (non-HBOC phenotype) (Table 4).

### 3.3. PALB2 Variant Characterization

Overall, 13 different disease-causing variants were identified in 22 patients among all the investigated cases (Table 2). Interestingly, no missense genetic alteration was detected among our patients, all P/LP variants lead to loss-of-function of the *PALB2* gene. The two most common variants were NM_024675.4:c.509_510delGA (p.(Arg170IlefsTer14)) and NM_024675.4:c.109-2A > G (p.?), which were detected in six (27%; 6/22) and three (3/22; 14%) cases, respectively (Table 2). These were also reported in the NCBI ClinVar database in several cases (by 38 and 4 submitters, respectively). Of the 13 P/LP *PALB2* variants, two were copy number variations affecting exons 9–10 and 11 as deletions (Table 2 and Figure 2).

In our cohorts, frameshift variants were more frequent compared to missense genetic alterations (Table 5). No statistically significant difference was found between the distribution of the *PALB2* P/LP mutation types regarding frameshift, stop, and missense variants in our samples compared to cases of the gnomAD non-cancer database, where the frameshift, stop, splice, and missense variant frequencies were 66.8%, 30.7%, 2.4%, and 0%, respectively (Table 5).

No correlation was identified between *PALB2* variant types and the age at tumor onset, gender, single or bilateral, or multiplex disease (Table 6).

## 4. Discussion

We analyzed the *PALB2* P/LP variant frequencies and mutational spectrum in Hungary for the first time. We investigated three cohorts of cancer patients: one as a set of consecutive patients with tumors characteristic of hereditary breast and ovarian cancer, one of young breast cancer patients (<33 years), and one cohort with non-HBOC tumors as disease controls. Additionally, we assessed the gnomAD non-cancer non-Finnish European population as a group of healthy (non-cancer) individuals. We found that the *PALB2* P/LP ratio was 1.4% (18/1280) among 1280 consecutive HBOC patients. This finding was similar to Canadian, British, and Hispanic populations [5,17,18,19,20]. The overall detection rate in the literature ranged between 0.36% and 4.8% [4,18,21,22,23,24,25], and the higher indices were observed in Finland, attributed to a founder mutation [22,26]. Low prevalence was observed in the Jewish Ashkenazi population, in Irish, Japanese, or Dutch studies [27,28,29,30,31,32]. In our study cohorts, the OR for breast cancer risk in women with *PALB2* P/LP variants was between 8.1 and 9.3 in breast cancer patients compared to non-HBOC cancer patients and non-cancer population, respectively.

Among young breast cancer patients, we detected P/LP *PALB2* variants in 1.5% (3/191), similar to studies by Cao (1.3%) and Sluiter (2%) [33,34]. Additionally, nearly significant differences (*p* = 0.0503) were seen in the probability of the age of tumor onset between *PALB2* P/LP carriers and patients with normal genotypes. Indeed, Zhou et al. found profoundly increased breast cancer risk for patients ≤ 30 years in the Chinese population compared to those > 30 years among *PALB2* P/LP variant carriers [25].

“Double mutations” (P/LP variant in more than one HBOC susceptibility gene) are rare [35], and in our study, these were observed in 13 cases (1%; 13/1280); however, none of them carried any *PALB2* disease-causing genetic alteration. Notably, transheterozygotes who have inherited deleterious mutations in both *BRCA1* and *BRCA2* were first reported in a Hungarian patient with breast/ovarian cancer [36]. Large-scale studies warrant uncovering of further transheterozygote pathogenic variants of HBOC genes and exploring the phenotype consequences of transheterozygosity.

While strong genotype–phenotype associations were not found in our study, probably due to the relatively low sample numbers, bilateral breast cancer, male breast cancer, and pancreatic cancer occurred among patients carrying P/LP *PALB2* variants. The more precise interrelation between these manifestations and *PALB2* P/LP variants should be further investigated on a larger, independent breast cancer cohort. The prevalence of male breast cancer in our cohort was comparable (MBC/FBC: 43/1237, ratio: 0.034) to those reported in other studies (MBC/FBC: 40/2893 [37], ratio: 0.013 and MBC/FBC: 419/9675, ratio: 0.043 [38]). We found a similar *PALB2* P/LP detection ratio between males (1/43; 2.3%) and females (21/1237; 1.6%). Still, for genetic counselors, it has to be considered that germline *PALB2* P/LP variants were also reported in males in other studies [4,18,39,40] and that it represents an increased risk for developing male breast cancer (odds ratio, OR = 6.6) [3,41]. Also, the risk for pancreatic cancer in *PALB2* heterozygote P/LP carriers is estimated to be 2–3% to age 80 years [3]. Regarding the bilateral/contralateral breast cancer risk, ACMG suggests that more systematic prospective data collection is needed to correctly address this question [3]. This year, however, a study including 15,104 prospectively followed women treated with ipsilateral surgery for invasive breast cancer reported a 35% 10-year cumulative incidence of contralateral breast cancer for *PALB2* truncating variant carriers with ER-negative breast cancer [42].

Among the two control groups (cancer patients with non-HBOC tumors and healthy control, non-cancer population), we found that the *PALB2* P/LP detection rate was low (0.088% and 0.076%, respectively). Also, this did not differ significantly from other HBOC susceptibility genes. This was in line with others’ findings, who also detected low frequency of *PALB2* P/LP variants as incidental/secondary findings in healthy controls [25,43]. However, for patients with incidentally identified *PALB2* variants surveillance programs according to NCCN guidelines [9,13] should be offered in order to early detection of potentially developing malignancies.

In the Hungarian HBOC population, NM_024675.4:c.509_510delGA and NM_024675.4:c.109-2A > G (p.?) were detected the most frequently, in 27% (6/22) and 14% (3/22), respectively. The c.509_510delGA seems to be a common variant reported in other populations as well [44]. To determine if it can be considered as a founder variant in the Hungarian population further studies are required. Janssen et al. reported the first three exons having the highest mutation rates (exon 1 (6.3%), exon 2 (6.7%), and exon 3 (5.8%)) [44]. In our study, P/LP variants were the most frequent in exon 4 (in 5 cases, 5/22, 23%). Among our patients, frameshift, nonsense, and splice variants were the most common, while P/LP missense genetic alterations were not observed. This was similar in the healthy control cohort and in the study of Weitzel et al., who detected a similar variant distribution [20]. This may be because a significantly larger number of loss-of-function mutations (e.g., frameshift, nonsense, splice, exonic deletions/duplications) have been reported as pathogenic/likely pathogenic since the functional validation of missense variants represents a greater challenge. For this, sophisticated functional assays such as protein-protein interaction or proficiency testing in homolog recombination repair should be applied. Still, a significant number of missense variants remains unclassified; therefore, ClinGen *PALB2* Variant Curation Expert Panel (VCEP) has made an effort to provide expert curation on P/LP *PALB2* variants [3,4].

*PALB2* studies have been mainly focused on truncating mutations, but the presence of variants of uncertain significance (VUS) has also been reported in patients [21,28,30,45,46], which represents a challenge for genetic counselors, clinicians, and patients as well. While we did not find differences between clinicopathological parameters of wild-type and *PALB2* VUS carriers, further functional characterization of *PALB2* VUSs will be able to discriminate some VUSs with pathogenic potential, hence aiding the clinical practice. Until the clarification of the role of VUSs, ACMG recommends that *PALB2* VUS are not used to guide clinical management [3].

The relatively small number of *PALB2* P/LP variant carriers represents a limitation in the assessment of genotype–phenotype associations and the potential additive effects of extrinsic factors such as smoking, alcohol consumption, personality type, hypertension, obesity, physical inactivity, and dietary habits on disease manifestation. Due to the low compliance of probands’ family members regarding genetic testing, the de novo rate or potential protective factors in parents cannot be assessed reliably. These should be evaluated on larger, independent breast cancer cohorts.

## 5. Conclusions

*PALB2* P/LP variants are not rare. A total of 18 patients were identified with disease-causing variants among 1280 Hungarian HBOC patients during a one-and-a-half-year period. The c.509_510delGA variant was the most common in the studied Hungarian patient population. We did not find a significant difference in the detection ratio of *PALB2* disease-causing variants according to age; however, the median age of tumor onset was lower in *PALB2* P/LP carriers versus wild-type patients (44 vs. 48 years). In our cohort, the OR for breast cancer risk in women with *PALB2* P/LP variants was between 8.1 and 9.3 compared to non-HBOC cancer patients and the non-cancer population, respectively. Triple-negativity was higher among P/LP carriers compared to patients with wild-type genotype and VUS carriers (40% vs. 20%); however, this did not reach statistical significance. In our patient cohort, no significant difference regarding multiplex tumor occurrence, estrogen positivity, and HER2 positivity was observed. The low rate of *PALB2* incidental finding was similar to *BRCA1* and *BRCA2*, suggesting higher penetrance compared to *ATM* and *CHEK2* genes.

*PALB2* testing is important because of the associated high cancer risk, and including patients carrying P/LP variants in preventive and screening programs can improve their life expectancy similarly to *BRCA1/2* carriers.

## Figures and Tables

**Figure 1 cancers-15-04350-f001:**
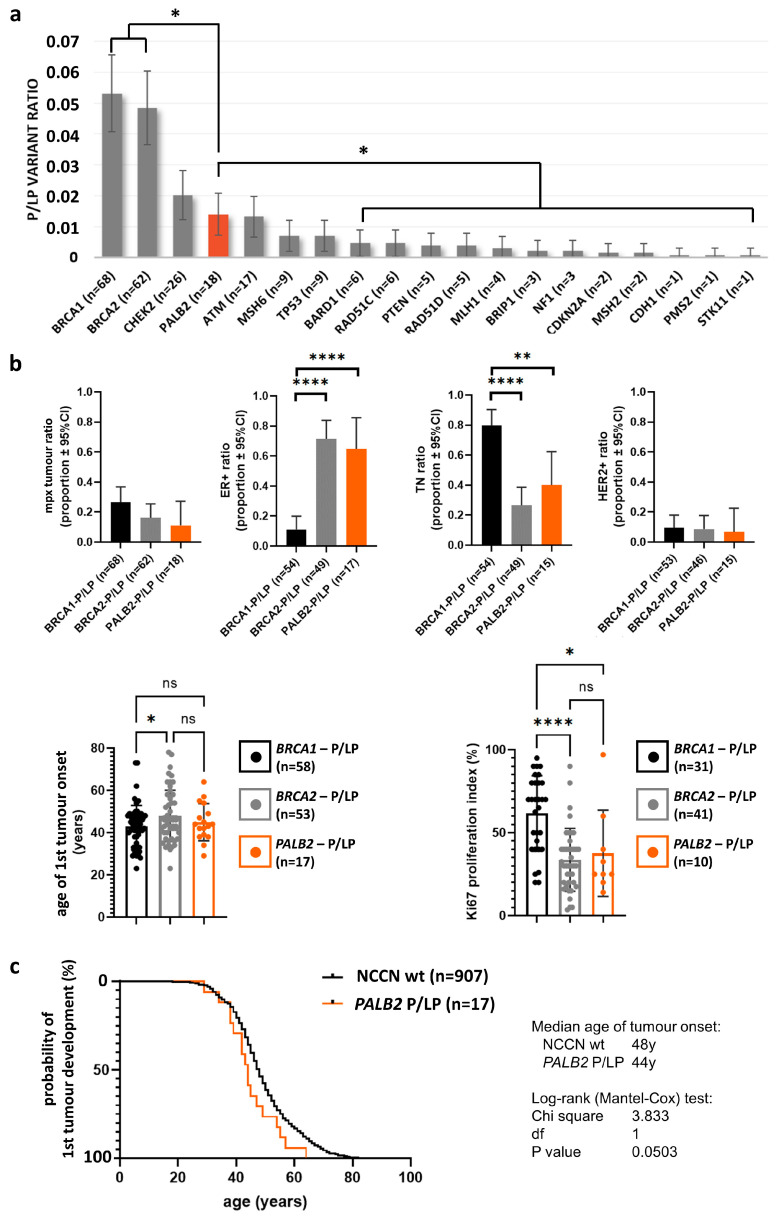
(**a**) Detection ratio of pathogenic/likely pathogenic variants (P/LP) in *PALB2* and other HBOC susceptibility genes. “n” represents the number of P/LP variants; (**b**) Clinicopathological characteristics of patients according to genotype. “n” represents the number of patients; (**c**) Age of first tumor diagnosis in wild-type and *PALB2* P/LP carriers. * *p* < 0.05; ** *p* ≤ 0.01; **** *p* ≤ 0.0001. NCCN wt: normal genotype (wild-type) of HBOC susceptibility genes indicated by the National Comprehensive Cancer Network Genetic/Familial High-Risk Assessment: Breast, Ovarian, and Pancreatic guideline; mpx: multiplex; ER +: estrogen receptor positive; TN: triple negative; HER2: human epidermal growth factor receptor 2; VUS: variant of uncertain significance.

**Figure 2 cancers-15-04350-f002:**
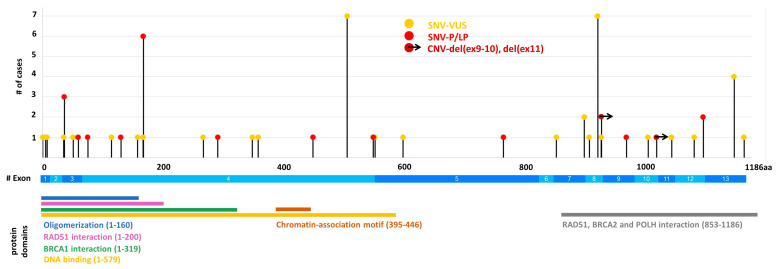
*PALB2* (pathogenic/likely pathogenic and VUS) variants detected in Hungarian patients. Domains are illustrated according to the Uniprot database (accessed on 10 July 2023). SNV: small nucleotide variant, CNV: copy number variant.

**Table 1 cancers-15-04350-t001:** Patient cohorts’ characteristics.

**HBOC cohort ***	
Gender	
Female (n)	1237
Male (n)	43
total (n)	1280
Age	
Average ± SD (years)	49 ± 11
Min–max (years)	18–84
Tumor types	
Breast cancer (n)	1145
Ovarian cancer (n)	89
Pancreatic cancer (n)	40
Prostate cancer (n)	30
**non-HBOC cohort ***	
Gender	
Female (n)	391
Male (n)	177
total (n)	568
Age	
Average ± SD (years)	42 ± 20
Min–max (years)	1–83
Tumor types	
HNPCC-related tumor types (colorectal, endometrial cancer) (n)	186
Endocrine-related cancer types (e.g., adrenal, pituitary, neuroendocrine tumors, thyroid cancer) (n)	213
Gastrointestinal (non-HNPCC-related) tumors (e.g., GIST, polyposis) (n)	3
other (rare/not classified/no syndrome-related) (n)	156
Multiplex tumors not fitting classical hereditary tumor syndromes (n)	10
**yBr cohort ***	
Gender	
Female (n)	191
Male (n)	0
total (n)	191
Age	
Average ± SD (years)	31 ± 2
Min–max (years)	18–33
Tumor types	
Single-sided breast cancer (n)	177
Bilateral breast cancer (n)	11
Multiplex tumors (n)	5

* HBOC cohort: hereditary breast and ovarian cancer; * non-HBOC cohort: oncology patients harboring non-HBOC tumor types; yBr cohort: young breast cancer patient cohort (<33 years); GIST: gastrointestinal stromal tumor; HNPCC: hereditary nonpolyposis colorectal cancer or Lynch syndrome; SD: standard deviation.

**Table 2 cancers-15-04350-t002:** Identified *PALB2* variants.

Variant Name (HGVSC) MANE NM_024675.4	Variant Name (HGVSP) MANE NP_078951.2	Variant_Type (Affected Exon/ All Exons)	Variant rs ID	Nr of Patients Carrying the Variant, Cohort	Own Assessment	ClinVar InterpRetation	Varsome Interpretation	Franklin Genoox Interpretation	ClinGen VCEP Interpretation
c.109-2A > G	p.?	splice_acceptor_(i2/12)	rs730881897	3, HBOC	P	LP	P	P	n.a.
c.172_175delTTGT	p.(Gln60ArgfsTer7)	frameshift_(e3/13)	-	1, HBOC	P	LP	P	P	n.a.
c.228_229delAT	p.(Ile76MetfsTer4)	frameshift_(e4/13)	-	1, HBOC	P	LP	P	P	n.a.
c.395delT	p.(Val132AlafsTer45)	frameshift (e4/13)	rs180177085	1, HBOC	P	P	P	P	n.a.
c.509_510delGA	p.(Arg170IlefsTer14)	frameshift_(e4/13)	rs515726123	4, HBOC; 2, yBr	P	P	P	P	n.a.
c. 886delA	p.(Met296Ter)	stop_gained_(e5/13)	-	1, yBr	P	P	P	P	n.a.
c.1369G > T	p.(Glu457Ter)	stop_gained_(e4/13)	rs878855099	1, HBOC	P	P	P	P	n.a.
c.1676_1677delAAinsG	p.(Gln559ArgfsTer2)	frameshift (e4/13)	rs515726073	1, HBOC	P	P	P	P	n.a.
c.2336C > A	p.(Ser779Ter)	stop_gained_(e5/13)	-	1, HBOC	LP	P/LP	LP	LP	n.a.
c.(2834 + 1_2835-1)_ (3113 + 1_3114-1)del	p.?	cnv-del_(e9-10/13)	-	2, HBOC	P	n.a.	n.a.	n.a.	n.a.
c.2959C > T	p.Gln987Ter	stop_gained_(e5/13)	-	1, non-HBOC	P	n.r.	LP	LP	n.a.
c.(3113 + 1_3114-1)_ (3201 + 1_3202-1)del	p.?	cnv-del_(e11/13)	-	1, HBOC	P	n.a.	n.a.	n.a.	n.a.
c.3350G > A	p.(Arg1117Lys)	missense_(e12/13)| splice_region	rs876659859	2, HBOC	P	P	LP	P	P
c.-44C > T	p.?	5_prime_UTR_(e1/13)	-	1, HBOC	VUS-	n.a.	LB	VUS-	n.a.
c.13C > T	p.(Pro5Ser)	missense_(e1/13)	rs377085677	1, HBOC	VUS	CONF	LB	VUS	n.a.
c.108 + 50A > G	p.?	intron (i2/12)	rs768185311	1, HBOC	VUS-	n.r.	LB	VUS-	n.a.
c.154G > A	p.(Val52Ile)	missense (e3/13)	rs876659444	1, HBOC	VUS	VUS	LB	VUS	n.a.
c.212-10delT	p.?	intron_(i3/12)	-	1, HBOC	VUS-	CONF	VUS	VUS-	n.a.
c.349C > A	p.(Pro117Thr)	missense_(e4/13)	rs1413238389	1, HBOC	VUS	VUS	LB	VUS	n.a.
c.481G > C	p.(Asp161His)	missense (e4/13)	rs769841151	1, HBOC	VUS	VUS	LB	VUS	n.a.
c.509G > C	p.(Arg170Thr)	missense (e4/13)	rs898765598	1, HBOC	VUS	VUS	LB	VUS	n.a.
c.814G > A	p.(Glu272Lys)	missense_(e4/13)	rs515726127	1, HBOC	VUS	VUS	LB	VUS	n.a.
c.1063T > G	p.(Leu355Val)	missense_(e4/13)	rs1555461473	1, HBOC	VUS	VUS	LB	VUS	n.a.
c.1093A > G	p.(Arg365Gly)	missense_(e4/13)	rs773001248	1, HBOC	VUS	VUS	LB	VUS	n.a.
c.1544A > G	p.(Lys515Arg)	missense_(e4/13)	rs515726072	7, HBOC	VUS	CONF	LB	VUS	n.a.
c.1685-52G > C	p.?	intron_(i4/12)	rs1221707621	1, HBOC	VUS-	n.r.	LB	VUS-	n.a.
c.1828A > T	p.(Thr610Ser)	missense_(e5/13)	-	1, HBOC	VUS	n.r.	LB	VUS	n.a.
c.2606C > G	p.(Ser869Cys)	missense_(e7/13)	rs779279139	1, HBOC	VUS	VUS	VUS	VUS	n.a.
c.2748 + 56_2748 + 58delAGA	p.?	intron_(i7/12)	rs753566712	2, HBOC	VUS-	n.r.	LB	VUS-	n.a.
c.2773G > C	p.(Val925Leu)	missense_(e8/13)	rs180177125	1, HBOC	VUS	VUS	LB	VUS	n.a.
c.2816T > G	p.(Leu939Trp)	missense_(e8/13)	rs45478192	7, HBOC	VUS	CONF	B	VUS-	n.a.
c.2834 + 68A > G	p.?	intron_(i8/12)	-	1, HBOC	VUS-	n.a.	LB	VUS-	n.a.
c.3071A > G	p.(Glu1024Gly)	missense_(e10/13)	-	1, HBOC	VUS	n.a.	VUS	VUS	n.a.
c.3191A > G	p.(Tyr1064Cys)	missense_(e11/13)	rs730881893	1, HBOC	VUS	VUS	VUS	VUS	n.a.
c.3306C > G	p.(Ser1102Arg)	missense_(e12/13)	rs515726112	1, HBOC	VUS	VUS	VUS	VUS	n.a.
c.3508C > T	p.(His1170Tyr)	missense_(e13/13)	rs200283306	4, HBOC	VUS	CONF	LB	VUS	n.a.
c.*33T > A	p.?	3_prime_UTR_(e13/13)	-	1, HBOC	VUS-	n.a.	LB	VUS-	n.a.

HGVSC: Sequence Variant Nomenclature on the cDNA level according to Human Genome Variation Society; HGVSP: Sequence Variant Nomenclature on the protein level according to Human Genome Variation Society; MANE: Matched Annotation from NCBI and EMBL-EBI; VCEP: ClinGen Variant Curation Expert Panel; P: pathogenic; LP: likely pathogenic; B: benign; LB: likely benign; VUS: variant of uncertain significance; CONF: conflicting; HBOC cohort: hereditary breast and ovarian cancer; non-HBOC cohort: oncology patients harboring non-HBOC tumor types; yBr cohort: young breast cancer patient cohort (<33 years); n.a.: not available; n.r.: not reported in NCBIClinVar database.

**Table 3 cancers-15-04350-t003:** Comparison and odds of *PALB2* P/LP variant among different cohorts.

	HBOC	NonHBOC	YHBOC	GnomAD-NonCancer
**HBOC**	-	**OR: 8.1;** **95CI:1.456 to 84.93;** ***p* = 0.0119**	OR: 0.89; 95CI:0.2926 to 2.897; *p* = 0.7470	**OR: 9.3;** **95CI:5.739 to 15.14;** ***p* < 0.0001**
**non-HBOC**		-	OR: 0.1105; 95CI: 0.008497 to 0.7470; ***p = 0.0512***	OR: 0.86; 95CI:0.1214 to 6.199; *p* = 0.8873
**yHBOC**			-	**OR: 10.43;** **95CI:3.306 to 32.90;** ***p* < 0.0001**
**gnomAD-nonCancer**				OR: 1.0 (reference)-

HBOC cohort: hereditary breast and ovarian cancer; non-HBOC cohort: oncology patients harboring non-HBOC tumor types; yBr cohort: young breast cancer patient cohort (<33 years). **Bold letters** indicate statistically significant results. ***Bold italic letters*** indicate statistically near-significant results.

**Table 4 cancers-15-04350-t004:** Detection ratio of *PALB2* P/LP as an incidental finding in non-HBOC patients and gnomAD non-cancer population.

Gene Name	Non-HBOC Cohort		GnomAD Non-Cancer Cohort		Chi-Squared Test		
	Detected Cases/All Cases	Allele Frequency	Detected Cases/All Cases	Allele Frequency	Difference	95% CI	*p*-Value
*PALB2*	1/568	0.00088	205/134,187	0.000764	0.0116%	−0.0617 to 0.4206	0.8877
*BRCA1*	2/568	0.001761	297/134,187	0.001107	0.0654%	−0.0631 to 0.5292	0.5088
*BRCA2*	2/568	0.001761	423/134,187	0.001576	0.0185%	−0.1103 to 0.4823	0.8754
*ATM*	4/568	0.003521	458/134,187	0.001707	0.1814%	−0.0343 to 0.7313	0.1403
*CHEK2*	5/568	0.004401	2406/134,187	0.008965	0.5464%	−0.1307 to 0.7110	0.1030

**Table 5 cancers-15-04350-t005:** Pathogenic/likely pathogenic *PALB2* variant types.

Variant Type	Current Study Cohort—Disease-Related Genetic Alterations			GnomAD Non-Cancer Cohort—Incidental Finding, Not Associating with Phenotype			Fisher’s Exact Two-Tailed *p*-Value
	Patient Nr Carrying Variant (n)	Nr of Variant Type/Nr of All P/LP Variants (22)	95% CI	Patient Nr Carrying Variant (n)	Nr of Variant Type/Nr of All P/LP Variants (205)	95% CI	
frameshift	10	0.454545	0.2691 to 0.6535	137	0.668293	0.6012 to 0.7292	0.3521
stop	4	0.181818	0.0671 to 0.3912	63	0.307317	0.2481 to 0.3736	0.4652
splice	5	0.3705	0.0971 to 0.4385	5	0.02439	0.0089 to 0.0574	0.0023 **
missense	0	0 *	0.0000 to 0.1755	0	0	0.0000 to 0.0221	1.0000
CNV	3	0.136364	0.0390 to 0.3418	-	n.a.	n.a.	n.a.

*: Fisher *p* = 0.0034 comparing frameshift vs. missense variant frequency. **: Fisher *p* = 0.0023 comparing P/LP *PALB2* splice variant frequency in our study samples vs. gnomAD non-cancer population. CNV: copy number variants; n.a.: not applicable.

**Table 6 cancers-15-04350-t006:** Patients’ phenotype according to different *PALB2* P/LP variants in different cohorts.

#	Gender	1st Tumor		2nd Tumor		P/LP *PALB2* Variant	Variant Type	Cohort *
		Type	Age at Onset	Type	Age at Onset			
1	F	Breast cancer	39	-	-	c.109-2A > G p.?	splice_acceptor	HBOC
2	F	Breast cancer	49	-	-	c.109-2A > G p.?	splice_acceptor	HBOC
3	F	Breast cancer	42	Breast cancer	60	c.109-2A > G p.?	splice_acceptor	HBOC
4	F	Breast cancer	42	-	-	c.172_175delTTGT p.(Gln60ArgfsTer7)	frameshift	HBOC
5	F	Breast cancer	54	-	-	c.228_229delAT p.(Ile76MetfsTer4)	frameshift	HBOC
6	F	Breast cancer	55	-	-	c.395delT p.(Val132AlafsTer45)	frameshift	HBOC
7	M	Breast cancer	57	-	-	c.509_510delGA p.(Arg170IlefsTer14)	frameshift	HBOC
8	F	Breast cancer	29	-	-	c.509_510delGA p.(Arg170IlefsTer14)	frameshift	HBOC
9	F	Breast cancer	44	-	-	c.509_510delGA p.(Arg170IlefsTer14)	frameshift	HBOC
10	F	Breast cancer	45	-	-	c.509_510delGA p.(Arg170IlefsTer14)	frameshift	HBOC
11	F	Breast cancer	44	-	-	c.1369G > T p.(Glu457Ter)	stop_gained	HBOC
12	F	Breast cancer	64	-	-	c.1676_1677delAAinsG p.(Gln559ArgfsTer2)	frameshift	HBOC
13	F	Pancreatic cancer	54	-	-	c.2336C > A p.(Ser779Ter)	stop_gained	HBOC
14	F	Breast cancer	38	-	-	c.3350G > A p.(Arg1117Lys)	missense|splice_region	HBOC
15	F	Breast cancer	38	-	-	c.3350G > A p.(Arg1117Lys)	missense|splice_region	HBOC
16	F	Breast cancer	47	-	-	c.(3113 + 1_3114-1)_ (3201 + 1_3202-1)del p.?	cnv-del	HBOC
17	F	Breast cancer	34	-	-	c.(2834 + 1_2835-1)_ (3113 + 1_3114-1)del p.?	cnv-del	HBOC
18	F	Breast cancer	43	Breast cancer	43	c.(2834 + 1_2835-1)_ (3113 + 1_3114-1)del p.?	cnv-del	HBOC
19	F	Breast cancer	30	-	-	c. 886delA p.(Met296Ter)	stop_gained	yBR
20	F	Breast cancer	32	Ovarian cancer	37	c.509_510delGA p.(Arg170IlefsTer14)	frameshift	yBR
21	F	Breast cancer	33	-	-	c.509_510delGA p.(Arg170IlefsTer14)	frameshift	yBR
22	F	Endometrial cancer	34	-	-	c.2959C > T p.Gln987Ter	stop_gained	non-HBOC (incidental)

* HBOC cohort: hereditary breast and ovarian cancer; non-HBOC cohort: oncology patients harboring non-HBOC tumor types; yBr cohort: young breast cancer patient cohort (<33 years).

## Data Availability

The datasets generated during and/or analyzed during the current study are presented in the current manuscript and are available from the corresponding author upon reasonable request.

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
