# Peer review of "PALB2* Variants Extend the Mutational Profile of Hungarian Patients with Breast and Ovarian Cancer"

_cancers, 2023, doi:10.3390/cancers15174350_

Round 1

Reviewer 1 Report

I am pleased to provide a review of this manuscript. The study delves into the variability of pathogenic/likely pathogenic (P/LP) variants of PALB2, a crucial breast cancer gene, across diverse populations. The authors meticulously examined three distinct cohorts, concurrently comparing data from 134,187 non-cancer individuals. The findings underscore the significance of PALB2 P/LP variants among breast and/or ovarian cancer patients, particularly prevalent in Hungarian patients. The study underscores the pivotal role of PALB2 testing in managing cancer risk and extends to potential implications for non-breast/ovarian cancer patients. Furthermore, this study's insights contribute to guiding the inclusion of PALB2 in routine molecular genetic testing for breast cancer patients. I don’t have any criticisms.

Author Response

Point-by-point response to reviewers

Reviewer 1:

I am pleased to provide a review of this manuscript. The study delves into the variability of pathogenic/likely pathogenic (P/LP) variants of PALB2, a crucial breast cancer gene, across diverse populations. The authors meticulously examined three distinct cohorts, concurrently comparing data from 134,187 non-cancer individuals. The findings underscore the significance of PALB2 P/LP variants among breast and/or ovarian cancer patients, particularly prevalent in Hungarian patients. The study underscores the pivotal role of PALB2 testing in managing cancer risk and extends to potential implications for non-breast/ovarian cancer patients. Furthermore, this study's insights contribute to guiding the inclusion of PALB2 in routine molecular genetic testing for breast cancer patients. I don’t have any criticisms.

Response:

We thank the Reviewer for his/her work reviewing our manuscript and we are grateful the Reviewer for his/her overall positive opinion about our work.

Reviewer 2 Report

PALB2 variants extend the mutational profile of Hungarian patients with breast and ovarian cancer

Review:

The article aims to investigate the prevalence, mutational spectrum, and clinical significance of PALB2 gene variants in the context of hereditary breast and ovarian cancer (HBOC). It seeks to explore PALB2's role as a key component in the DNA repair pathway, particularly in forming a tri-molecular complex with BRCA1 and BRCA2 for effective homologous DNA recombination. The study's primary objectives include analyzing PALB2 variant frequencies, assessing genotype-phenotype associations, and providing insights for clinical recommendations. The results reveal that PALB2 pathogenic/likely pathogenic (P/LP) variants contribute to an increased risk of breast cancer, positioning PALB2 as the third most significant breast cancer gene following BRCA1 and BRCA2. The study identifies specific PALB2 P/LP variants prevalent in the Hungarian population and highlights the potential implications for cancer prevention and management strategies. The study's findings underscore the significance of PALB2 in HBOC, however, it has also raised specific questions that require addressing to enhance the clarity and comprehensiveness of the research.

Minor comments and questions that require addressing:

1.     What was the rationale behind including male subjects in the HBOC study? If there was a specific scientific justification, why were the numbers of male subjects relatively low?

2.     Regarding the samples that tested positive for PALB2 P/LP variants within the Non-HBOC group, could you please provide information about the associated tumor type for those samples and offer a discussion on potential biological insights?

3.     Is there any association identified between PALB2 variants and extrinsic factors such as smoking, alcohol consumption, personality type, hypertension, obesity, physical inactivity, and dietary habits?

4.     Why was the GRCh37 genome used, considering that Dragen 4.0 is capable of conducting analyses on GRCh38? and What quality parameters were assessed during the analysis of the sequencing data?

5.     Regarding these variants, were they de novo, inherited, or a combination of both? If inherited, could you please discuss the molecular connections with any potential protective factors in parents?

6.     Among the 18 HBOC subjects, how many overlapped with the 235 samples that were detected with disease-causing variants in at least one of the breast cancer susceptibility genes? Furthermore, if these overlaps extended to genes other than BRCA1 and BRCA2, what potential molecular connections might be inferred?

7.     As mentioned in line 194, only one PALB2 pathogenic/likely pathogenic (P/LP) variant overlapped with the 235 disease-causing variant-associated subjects. Additionally, considering the well-established interaction between PALB2, BRCA1, and BRCA2, did the authors examine the levels of BRCA1 and BRCA2 genes to assess the potential impact of PALB2 variants in the subset of 18 or 17 HBOC subjects?

Author Response

Point-by-point response to reviewers

Reviewer 2:

The article aims to investigate the prevalence, mutational spectrum, and clinical significance of PALB2 gene variants in the context of hereditary breast and ovarian cancer (HBOC). It seeks to explore PALB2's role as a key component in the DNA repair pathway, particularly in forming a tri-molecular complex with BRCA1 and BRCA2 for effective homologous DNA recombination. The study's primary objectives include analyzing PALB2 variant frequencies, assessing genotype-phenotype associations, and providing insights for clinical recommendations. The results reveal that PALB2 pathogenic/likely pathogenic (P/LP) variants contribute to an increased risk of breast cancer, positioning PALB2 as the third most significant breast cancer gene following BRCA1 and BRCA2. The study identifies specific PALB2 P/LP variants prevalent in the Hungarian population and highlights the potential implications for cancer prevention and management strategies. The study's findings underscore the significance of PALB2 in HBOC, however, it has also raised specific questions that require addressing to enhance the clarity and comprehensiveness of the research.

Response: We thank the Reviewer for his/her accurate and detailed review, and we are grateful for the overall positive opinion about our work.

Minor comments and questions that require addressing:

  1. What was the rationale behind including male subjects in the HBOC study? If there was a specific scientific justification, why were the numbers of male subjects relatively low?

Response: The main cohort consisted of consecutive breast and ovarian cancer patients referred to our department for genetic counselling and genetic testing. According to NCCN guideline (1) and the the Hungarian Ministry of Human Resources' professional healthcare guideline on genetic counselling (2), male breast cancer patients at any age should be considered for testing of high-penetrance breast cancer susceptibility genes (specifically, BRCA1, BRCA2, CHD1, PALB2, PTEN and TP53). Therefore, in this prospective study between September 2021 and March 2023, we analysed genetic data of consecutively submitted breast cancer patient regardless of their gender.

We agree with the Reviewer, that the number of male breast cancer patients can be considered relatively low (among the consecutively referred breast cancer patients 43 were males (MBC) vs. 1237 females (FBC)). However, in another recent study published in Scientific Reports by Scomersi et al. (3) 40 MBC were analysed against 2893 FBC cases which were prospectively collected between January 2004 and May 2019 at the Breast Unit of University Hospital of Trieste. This represent 3x less MBC/FBC ratio compared to our study. Therefore, in a small country like Hungary with an approximately 10 million inhabitants, 43 male breast cancer cases collected in 18 months can be considered representative for our population. Higher sample numbers can be obtained by consortia collaboration, like in the manuscript by Silvestri et al. (4), analysing the data of 419 MBC and 9675 FBC patients. Our MBC/FBC ratio (43/1237; 0.034) was similar to their study (419/9675; 0.043).

Among our male breast cancer patients PALB2 pathogenic/likely pathogenic variant was detected only in one case. The PALB2 P/LP detection ratio was not statistically different between males (1/43; 2.3%) and females (21/1237; 1.6%), but this should be further validated on an extended sample size. Despite the relatively low number of our male breast cancer cases, these data may contribute to public knowledge in the near future when P/LP PALB2 variant detection ratio and genotype-phenotype associations will be evaluated on independent male breast cancer patient cohorts.

References:

  1. Daly M.B. et al. Genetic/Familial High-Risk Assessment: Breast, Ovarian, and Pancreatic, Version 2.2021, NCCN Clinical Practice Guidelines in Oncology. J Natl Compr Canc Netw. 2021 Jan 6;19(1):77-102. doi: 10.6004/jnccn.2021.0001. PMID: 33406487.
  2. The Ministry of Human Resources' professional healthcare guideline on genetic counseling. No. 20 of 2020. EüK., effective from 14th December 2020. http://www.hbcs.hu/uploads/jogszabaly/3278/fajlok/2020_EuK_20_szam_EMMI_szakmai_iranyelv_2.pdf
  3. Scomersi, S. et al. Comparison between male and female breast cancer survival using propensity score matching analysis. Sci Rep 11, 11639 (2021). https://doi.org/10.1038/s41598-021-91131-4
  4. Silvestri, V. et al. Male breast cancer in BRCA1 and BRCA2 mutation carriers: pathology data from the Consortium of Investigators of Modifiers of BRCA1/2 . Breast Cancer Res 18, 15 (2016). https://doi.org/10.1186/s13058-016-0671-y

Please find this clarification and additional information highlighted: page 3, line 100-106; page 5, line 176-179; page 11-12, line310-318 and 363-368.

  1. Regarding the samples that tested positive for PALB2 P/LP variants within the Non-HBOC group, could you please provide information about the associated tumor type for those samples and offer a discussion on potential biological insights?

Response:

In the non-HBOC control group, only one patient was identified with disease-causing PALB2 variant. This particular patient had endometriosis at the age of 34 years. The genetic testing was indicated because her mother and grandmother had breast cancer at the age of 45 and 60 years, respectively, however both of them were unavailable for testing. Due to the young age of this PALB2 P/LP variant carrier, PALB2-related tumour types could not be excluded in the future.

Although incidentally identified PALB2 P/LP variants are rare, but for these carriers, surveillance program according to NCCN guideline (NCCN Clinical Practice Guidelines Genetic/Familial High-Risk Assessment: Breast, Ovarian and Pancreatic 2.2023) should be offered in order to early detection of potentially developing malignancies.

Please find this additional information highlighted: page 8, line 226-230 and page 12, line 334-336

  1. Is there any association identified between PALB2 variants and extrinsic factors such as smoking, alcohol consumption, personality type, hypertension, obesity, physical inactivity, and dietary habits?

Response:

We thank the Reviewer for drawing our attention to the assessment of life-style parameters versus genetics. During pre-test genetic consultation, data regarding alcohol consumption, smoking habits, and body mass index were collected. We compared this information to data obtained by the Hungarian Central Statistical Office during the European population health survey in 2019 in general Hungarian population. We found that smoking was more frequent among Hungarian female breast cancer patients with PALB2 P/LP variant (70%) compared to Hungarian control population (38,5%). We did not find any difference regarding alcohol consumption or body mass index between the two groups. However, because of the small sample number of PALB2 P/LP carriers, this comparison cannot be considered statistically truthful as the statistical power of this comparison did not reach 80% with a type I/II error rate alpha 0.05. Therefore, we omitted to present these results in the revised manuscript. Also, we included this information in the limitations of the study. Please find highlighted: page 12, line 363-368

  1. Why was the GRCh37 genome used, considering that Dragen 4.0 is capable of conducting analyses on GRCh38? and What quality parameters were assessed during the analysis of the sequencing data?

Response:

We agree with the reviewer, Dragen is indeed capable of mapping to GRCh38, however the older version (GRCh37) is still more widely used in clinical genetics. While, the most significant improvements in GRCh38 is the annotation of the centromere regions, GRCh38 is not yet 'finished', regional fixes/patches are released periodically. GRCh37 is still richer in annotations, and there are some studies showing discordant SNV calls between the two assemblies (e.g. https://bmcbioinformatics.biomedcentral.com/articles/10.1186/s12859-019-2620-0), but it can be concluded that these have low quality scores and mainly appear in genomic regions with low coverage. Also, existing literature most commonly reports GRCh37 coordinates, this allows easy comparison of our data. Thus, we consider using GRCh37 as currently it represents a more stable solution. Nevertheless, we routinely cross-map all variants to the newer assembly for our own in-lab data warehouse as well.

Therefore, we believe that in case of HBOC associated genes, using GRCh37 as reference has not a significant impact on variant calling. Additionally, in our study, all P/LP and VUS variants were validated by traditional Sanger sequencing and MLPA as well, and functional effect was evaluated during variant interpretation based on published literature, databases (e.g. ClinVar), computational predictions and cross-referencing patient phenotype.

Regarding quality parameters we used Illumina Dragen Germline Enrichment pipeline (v.4.0.3) with default settings about which details are available at: https://support-docs.illumina.com/APP/AppGermline-v403/Content/APP/AppGermline.htm.

In addition to built-in Dragen default quality cutoffs for both fastq generation, mapping and variant calling (which are all assessed automatically and are part of the final quality score of the variant calls), additional quality parameters provided by the Dragen software (more specifically, by the Enrichment App used for these analyses) were listed, archived and assessed. These included (among others): unique read/base enrichment, number of total and unique aligned reads, percent of duplicate aligned reads, percent Q30 bases, percent aligned bases, percent target coverage at 1X-10X-15X-20X-50X, percent callability, fragment length parameters (min-max-median-SD, number/Het-Hom ratio/Ts-Tv ratio of SNVs. These parameters are included in the multiqc reports generated by the software for each sample in pdf and html formats

Briefly, in nucleotide detection, correct nucleotide read with higher than Phred quality score 30 was accepted per position. In mapping metrics, input read number was average 2.5 million/sample and average % proper reads was 95. In variant calling, variant allele frequency (VAF) between 30-70% was accepted for heterozygosity. Average base coverage for HBOC genes was 208 (min: 76, max: 594 reads/base). Low covered bases (<10 reads/bp) represented an average 0.1% per genes.

We included this information to the revised manuscript, please find highlighted: page 4, line 140-147.

  1. Regarding these variants, were they de novo, inherited, or a combination of both? If inherited, could you please discuss the molecular connections with any potential protective factors in parents?

Response:

Despite our effort during post-test counselling, willingness in patients’ families for cascade testing is not very high, especially in case of the parents due to their “old” age. Of the 19 PALB2 P/LP carrier probands, family members showed up only in 37% (7/19). In families when parents were available for genetic testing, PALB2 variants were all inherited. However, due to the small sample number we cannot draw conclusions regarding variant de novo rate or potential protective factors in parents reliably.

We described this limitation in the revised manuscript, please find highlighted: page 12, line 363-368.

  1. Among the 18 HBOC subjects, how many overlapped with the 235 samples that were detected with disease-causing variants in at least one of the breast cancer susceptibility genes? Furthermore, if these overlaps extended to genes other than BRCA1 and BRCA2, what potential molecular connections might be inferred?

Response:

Of the 1280 consecutively investigated HBOC patients, 235 had P/LP variant in any of the HBOC-associated genes (including PALB2). Out of the 235 cases, 18 patients had P/LP PALB2 variant (so the identified 18 patients represent a set of the 235 cases). We did not observe any patients having both PALB2 and BRCA1/2 P/LP variants or PALB2 P/LP variants together with P/LP variants in any other HBOC-associated gene.

“Double mutation” (P/LP variant in more than one HBOC susceptibility gene) was observed in 13 cases, however, none of them involved any PALB2 disease-causing genetic alteration. These were most commonly associations of BRCA1/2 with a moderate penetrance gene such as CHEK2, ATM or MSH6. In these cases, the exact molecular connections should be further investigated, as to date, there are not many cases or series of patients that describe the co-occurrence of two pathogenic variants, especially with the combination of one high-risk and one moderate risk genes; and only case reports are available (Andrés et al.). Therefore, no clinical evidence has been established. Literature (consortial) data suggests that transheterozygotes who have inherited deleterious mutations in two high risk genes for breast cancer the clinical phenotypes resemble single heterozygotes (Rebbeck et al.). Nonetheless, both variants have to be considered in the carrier individual and during family screening as well.

Please find the clarification highlighted: page 5, line 179-181

References:

Andrés R et al. Double heterozygous mutation in the BRCA1 and ATM genes involved in development of primary metachronous tumours: a case report. Breast Cancer Res Treat. 2019;177(3):767-770

Rebbeck T.R et al. Inheritance of deleterious mutations at both BRCA1 and BRCA2 in an international sample of 32,295 women. Breast Cancer Res. 2016;18(1):112.

  1. As mentioned in line 194, only one PALB2 pathogenic/likely pathogenic (P/LP) variant overlapped with the 235 disease-causing variant-associated subjects. Additionally, considering the well-established interaction between PALB2, BRCA1, and BRCA2, did the authors examine the levels of BRCA1 and BRCA2 genes to assess the potential impact of PALB2 variants in the subset of 18 or 17 HBOC subjects?

Response:

We apologize for the inaccurate phrasing, that sentence was aimed to refer to PALB2 variants of uncertain significance (VUSs). We clarified the above-mentioned sentence as:

„ Out of the 235 positive HBOC patients, 9 carried a PALB2 variant of uncertain significance (VUS) in addition to a P/LP variant in any of the other HBOC-associated genes.” (Please find highlighted: page 8, line 207-209)

Only PALB2 VUSs were detected together with P/LP variants in HBOC-associated genes (no PALB2 P/LP variants was identified together with other P/LP variants in HBOC associated genes).

Although the potential interaction of the effect of PALB2 VUSs with BRCA1/2 is a relevant research question, we did not perform functional studies or gene expression experiment as VUSs currently have no clinical significance.

We would like to thank the Reviewer for his/her questions and suggestions, we hope that our responses and the revised manuscript will be acceptable.
